# Elevating acetyl-CoA levels reduces aspects of brain aging

**Antonio Currais[1]\*, Ling Huang[2], Joshua Goldberg[1], Michael Petrascheck[3], Gamze Ates[1], António Pinto-Duarte[4], Maxim N Shokhirev[2], David Schubert[1], Pamela Maher[1]\***

[1]Cellular Neurobiology Laboratory, The Salk Institute for Biological Studies, La Jolla, United States; [2]The Razavi Newman Integrative Genomics and Bioinformatics Core, The Salk Institute for Biological Studies, La Jolla, United States; [3]Department of Molecular Medicine, The Scripps Research Institute, La Jolla, United States; [4]Computational Neurobiology Laboratory, The Salk Institute for Biological Studies, La Jolla, United States

**Abstract** Because old age is the greatest risk factor for dementia, a successful therapy will require an understanding of the physiological changes that occur in the brain with aging. Here, two structurally distinct Alzheimer's disease (AD) drug candidates, CMS121 and J147, were used to identify a unique molecular pathway that is shared between the aging brain and AD. CMS121 and J147 reduced cognitive decline as well as metabolic and transcriptional markers of aging in the brain when administered to rapidly aging SAMP8 mice. Both compounds preserved mitochondrial homeostasis by regulating acetyl-coenzyme A (acetyl-CoA) metabolism. CMS121 and J147 increased the levels of acetyl-CoA in cell culture and mice via the inhibition of acetyl-CoA carboxylase 1 (ACC1), resulting in neuroprotection and increased acetylation of histone H3K9 in SAMP8 mice, a site linked to memory enhancement. These data show that targeting specific metabolic aspects of the aging brain could result in treatments for dementia.

**\*For correspondence:**
acurrais@salk.edu (AC);
pmaher@salk.edu (PM)

## Introduction

Age is the greatest risk factor for Alzheimer's disease (AD) and related dementias. While genetic risk factors have been the major focus of AD drug discovery, it is important to consider the progressively detrimental metabolic processes that take place with normal aging as possible therapeutic targets. Clinical evidence shows that a decline in cerebral glucose metabolism (hypometabolism) precedes the pathology and symptoms of AD and is more severe than that observed in normal aging (*Costantini et al., 2008*; *Cunnane et al., 2011*; *Yin et al., 2014*). Given that energy production from glucose supports the majority of brain activity as well as the maintenance of cellular homeostasis, it is likely that a failure to supply cells with adequate energy contributes to the neuropathological cascade in both aging and age-associated dementias such as AD (*Caldwell et al., 2015*).

Most of the energy derived from glucose oxidation is produced in mitochondria. Not surprisingly, a number of mitochondrial-dependent functions are found to be impaired during aging and AD (*Caldwell et al., 2015*; *Chan, 2006*; *Currais, 2015a*; *Lin and Beal, 2006*; *Onyango et al., 2016*; *Swerdlow and Khan, 2004*; *Yin et al., 2016*). In fact, brain mitochondrial dysfunction has been hypothesized to be responsible for the pathological hallmarks of AD (*Swerdlow and Khan, 2004*; *Yin et al., 2016*).

Because the current approach to AD drug discovery has largely failed, we devised a novel drug discovery paradigm based on phenotypic screening assays that mimic numerous aspects of old age-associated neurodegeneration and brain pathology, including energy failure and mitochondrial dysfunction (*Prior et al., 2014*). We have identified two compounds that are very neuroprotective –

CMS121 and J147. J147 is active in transgenic AD animal models (*Chen et al., 2011*; *Daugherty et al., 2017*; *Prior et al., 2013*; *Prior et al., 2016*). It enhances memory and prevents some aspects of aging in rapidly aging (15 month median lifespan) senescence-accelerated prone 8 (SAMP8) mice when administered early in life (*Currais et al., 2015b*). CMS121 is a more potent derivative of the flavonol fisetin that maintains most of its biological properties (*Chiruta et al., 2012*). CMS121 is not as well studied as J147, but we have recently shown that fisetin is able to ameliorate some aspects of aging in SAMP8 mice (*Currais et al., 2018*). The target of J147 is the alpha subunit of ATP synthase, which engages a neuroprotective response involving the activation of AMP-activated protein kinase (AMPK) (*Goldberg et al., 2018*). ATP synthase is an anti-aging target in *C. elegans* (*Chin et al., 2014*). We have ruled out that CMS121 also targets ATP synthase (unpublished data), and its molecular targets are currently under investigation. However, because the two compounds were developed based upon brain toxicities associated with aging and therefore share similar biological activities in vitro, we hypothesized that they may mitigate some aspects of aging brain metabolism and pathology via a common pathway despite differences in molecular structure and direct targets.

To test this idea, we fed CMS121 and J147 to aged SAMP8 mice and used a multiomics approach to identify modes of action. We first show that both compounds reduce metabolic and gene transcription markers of aging in the SAMP8 model of aging and dementia when administered at a late stage of the aging process. We further demonstrate that both compounds share a mechanism of action that maintains high levels of acetyl-coenzyme A (acetyl-CoA), at least in part, by the inhibition of acetyl-CoA carboxylase 1 (ACC1). Importantly, the compounds increase histone acetylation in cultured neurons and SAMP8 mice at a site on histone H3 that is required for memory formation (*Mews et al., 2017*). Together, these data show that aging and dementia share a common metabolic pathway related to brain mitochondrial function that can be therapeutically targeted.

## Results

### Aging is associated with changes in the hippocampal transcriptome that are prevented by CMS121 and J147

To identify age-dependent changes in brain metabolism that are causally associated with dementia, we tested CMS121 and J147 in SAMP8 mice. The SAMP8 mice are a model of accelerated aging that develop a progressive, age-associated decline in brain function as well as a number of brain pathologies similar to human dementia and AD patients (*Akiguchi et al., 2017*; *Butterfield and Poon, 2005*; *Currais et al., 2015b*; *Morley et al., 2012*; *Pallas et al., 2008*; *Takeda, 2009*). Some of the pathological traits developed by the SAMP8 mice with aging that are also found in AD include: progressive decline in brain function with early deterioration in learning and memory; increased oxidative stress; inflammation; vascular impairment; gliosis; Aβ accumulation and tau hyperphosphorylation (*Cheng et al., 2014*; *Morley et al., 2012*; *Takeda, 2009*).

We asked whether J147 and CMS121 could prevent the development of the aging phenotype in SAMP8 mice when administered in the last quadrant of their lifespan, reflecting the therapeutic context for human AD patients. Treatment with the diets containing the compounds started at 9 months of age and continued for four months (13 months of age); the median lifespan of these mice is 15 months.

We began our analyses with an unbiased approach by studying the entire transcriptome of brain hippocampal tissue. Principle component analysis (PCA) showed a clear distinction in the variation of the transcriptome between the 9 and 13 months old SAMP8 groups, indicating that aging has a strong effect on gene expression (*Figure 1A*). Importantly, both compound-treated groups presented a distribution in the two major principle components (PCs) that was closer to the 9 months old mice, suggesting that the compounds can prevent transcriptomic changes that take place with aging (*Figure 1A*). This effect was particularly noticeable with CMS121. Transcriptomic drift analysis, a method to characterize the aging process that measures changes in gene expression at a global level (*Rangaraju et al., 2015*), showed a significant age-associated drift in gene expression between 9 and 13 months that was also suppressed by both compounds (*Figure 1B*).

To support the PCA and drift analysis, we examined the effects of the compounds on the expression of individual genes. 485 differently expressed (DE) genes were found between the 9 and 13

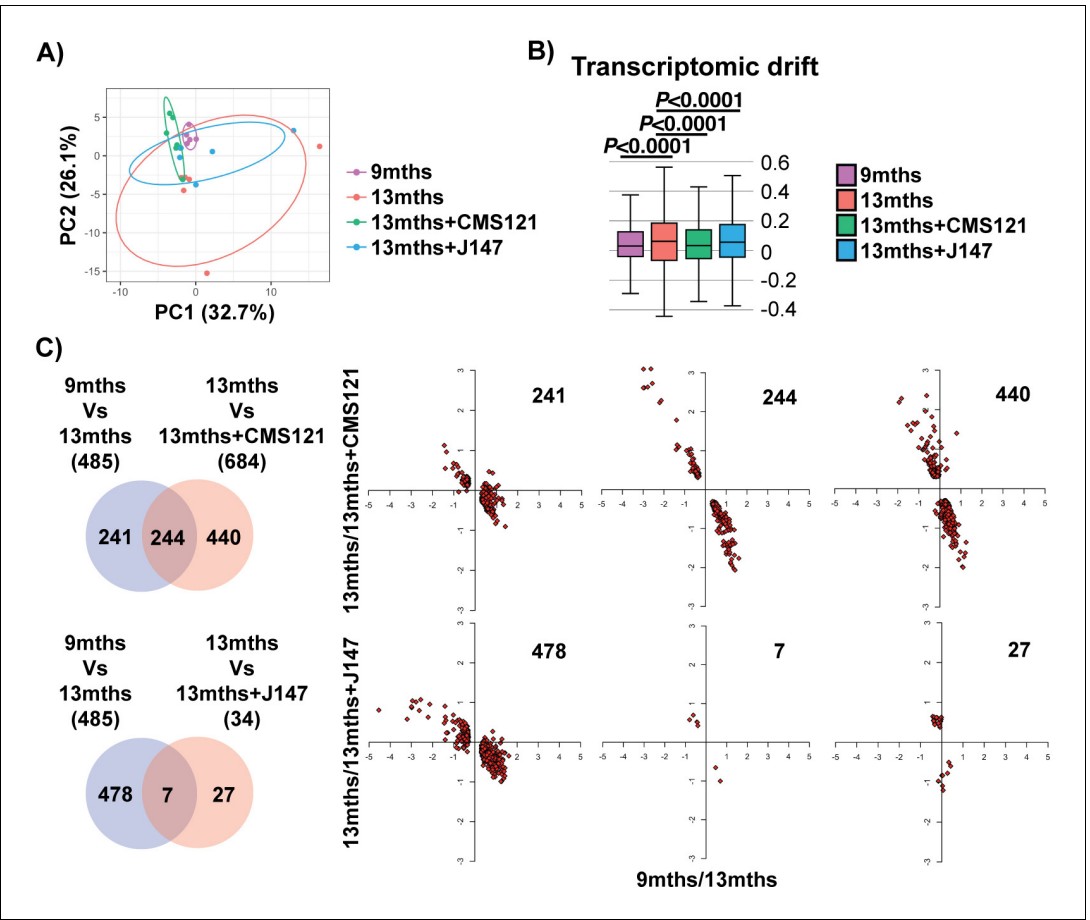

**Figure 1.** Aging of the hippocampal transcriptome in SAMP8 mice is significantly prevented by CMS121 and J147. (A) PCA of the top 10% most highly expressed genes in the hippocampus of 9 months, 13 months, 13 months +CMS121 and 13 months+J147 SAMP8 mice. Cutoff was sum(log2(FPKM+5))>=110.41 (2458 genes selected; n = 5–6/group). Ellipses show the 70% confidence interval. (B) Drift analysis of the whole transcriptome. Values are expressed as box-and-whisker plots. Brown-Forsythe test. n = 5–6/group. (C) Venn diagrams illustrating shared and uniquely affected genes between the 9 months vs 13 months old mice and 13 months vs each of the 13 months+compounds. n = 5–6/group. Please see Supplemental Materials and methods for detailed statistical analysis. Correlation of gene expression for each group of genes is represented on the right side of the panel (units are -log(fold change)).

months old SAMP8 mice (*Figure 1C* and *Supplementary file 1*). 684 and 34 DE genes were detected between the 13 months and 13 months+CMS121 and 13 months+J147 SAMP8 mice, respectively (*Figure 1C* and *Supplementary file 1*). Venn diagrams highlighting the shared DE genes between the 9 months vs 13 months old mice and 13 months vs each of the 13 months+compounds are shown in *Figure 1C*. Fold changes of the DE genes were log-transformed and plotted on correlation graphs for each pair of comparisons (*Figure 1C*, right side). For all of the DE genes that overlap between aging and compound treatment (shared DE genes), the age-dependent changes in expression were prevented by the compounds (*Figure 1C*, middle set of graphs). Furthermore, analysis of the DE genes that are specific for each comparison (those that do not overlap) also shows that the age-dependent alterations in most of these DE genes follow the same trend of prevention by the compounds. Thus, these data provide evidence that both compounds reduce aging at the level of the transcriptome in the hippocampus of SAMP8 mice when administered at a late stage of the aging process.

## CMS121 and J147 improve cognitive function in SAMP8 mice when administered at advanced stages of dementia

We next determined if the changes in gene transcription in old SAMP8 mice and the effects by CMS121 and J147 were associated with functional alterations in cognition.

Assessment of spontaneous behavior in the open field assay showed only a decline in average velocity between the 9 and 13 months SAMP8 mice (*Figure 2—figure supplement 1*). The compounds had no effect on this parameter or on body weights (*Figure 2—figure supplement 1*). To investigate the effects of CMS121 and J147 on age-associated cognitive decline, mice were tested using the elevated plus maze (*Figure 2A*) and the Barnes maze (*Figure 2B*).

The elevated plus maze examines disinhibition behavior based on the aversion of normal mice to open spaces. Dementia is clinically associated with disinhibition and AD mouse models tend to exhibit increased disinhibition (*Currais et al., 2015b*; *Prior et al., 2013*). 13 months old SAMP8 mice spent significantly more time in the open arms compared to 9 months old SAMP8 mice (*Figure 2A*). The age-associated increase in disinhibition was prevented by CMS121 and J147.

The Barnes maze is used to analyze spatial learning and hippocampal-dependent memory. In this assay, mice use visual cues to locate a hidden box. No changes between the groups were found in the escape latencies during the learning and the retention phases (data not shown). However, when tested during the reversal phase, which is more sensitive to deficits in memory and learning, CMS121 significantly improved the learning of the new location (*Figure 2B*). J147 had no significant effect. These data show that the transcriptomic aging in old SAMP8 mice is associated with a decline in cognitive function and that J147 and CMS121 prevent some of this deterioration.

## Suppression of transcriptomic aging by the compounds is specifically associated with the maintenance of brain mitochondria-related genes

We next asked whether there were specific cellular processes that were targeted by the compounds that could explain their beneficial effects. To address this, we carried out pathway analysis with the 485 DE genes that were altered with aging (*Figure 3A*) and with the 684 DE genes that were altered by CMS121 (*Figure 3B*). Given the small number of 34 DE genes detected with J147, this comparison was left out from the pathway analysis. However, the effects of J147 were included in the subsequent analyses.

CMS121 affected a number of different pathways (*Figure 3B*), some of which were also modified with aging (*Figure 3A*). The top KEGG pathway targeted by both aging and CMS121 was oxidative phosphorylation (OP). Moreover, the majority of the DE genes in this pathway are the same for both

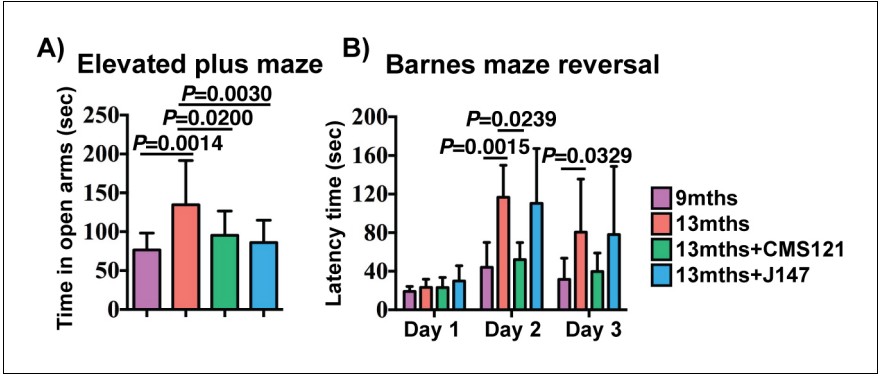

**Figure 2.** CMS121 and J147 preserve cognitive parameters in SAMP8 mice when administered at advanced stages of the phenotype. (**A**) The elevated plus maze was used to measure disinhibition behavior in 9 months, 13 months, 13 months+CMS121 and 13 months+J147 SAMP8 mice. One-way ANOVA followed by Tukey-Kramer post-hoc test (n = 11–18/group). (**B**) Spatial learning/memory was evaluated in the same mice by the Barnes maze assay. All data are mean ± SD. Two-way repeated measures ANOVA and post hoc Bonferroni corrected t-test (n = 5–9/group). The online version of this article includes the following figure supplement(s) for figure 2:

**Figure supplement 1.** Effect of CMS121 and J147 on activity parameters.

**Figure supplement 2.** Body weights of SAMP8 mice fed with vehicle, CMS121 and J147 diets.

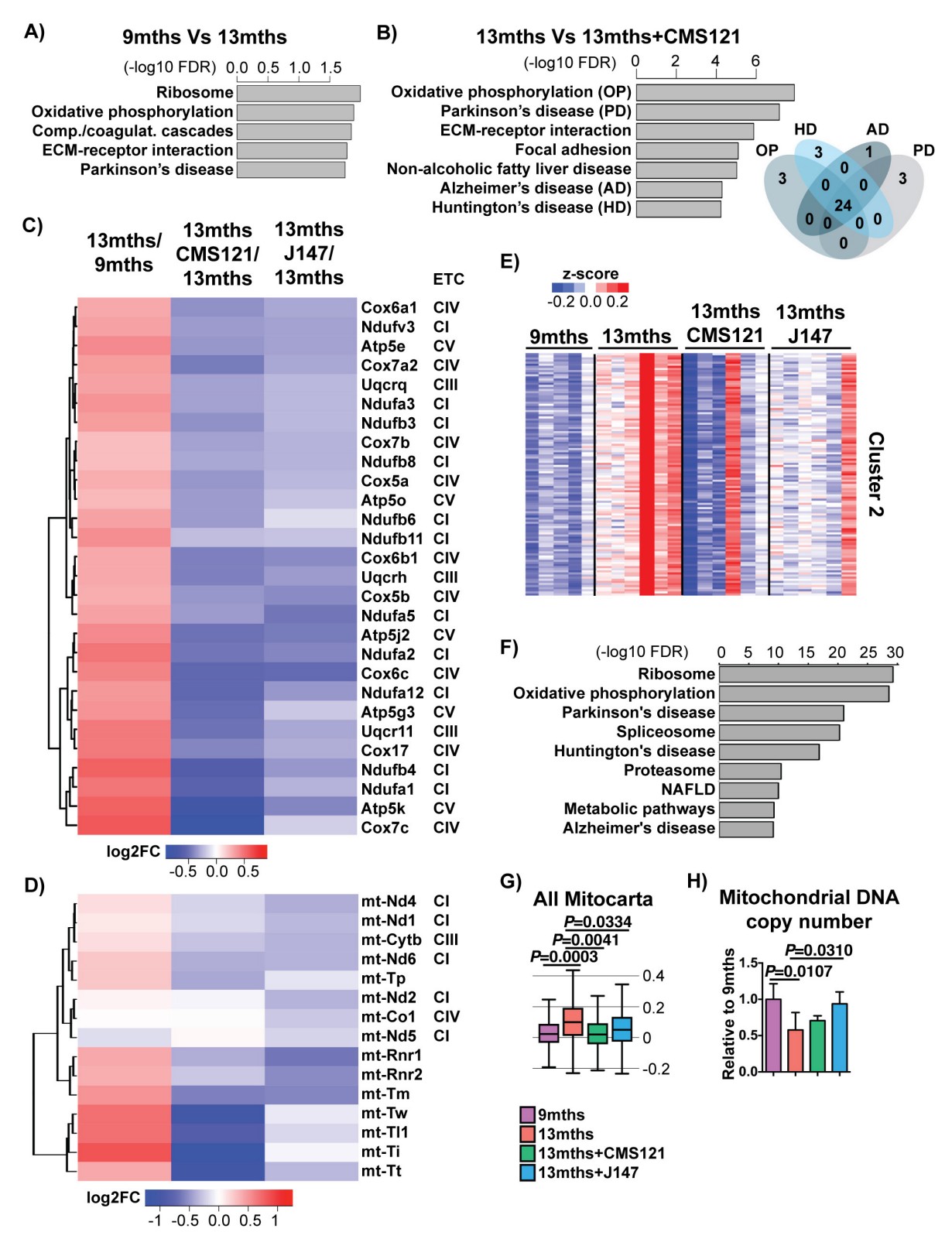

**Figure 3.** CMS121 and J147 specifically maintain the expression of genes associated with the mitochondria that are altered with aging in SAMP8 mice. Top KEGG pathways and respective enrichment scores associated with the 485 genes with altered expression in 9 months vs 13 months SAMP8 mice (**A**) and with the 684 genes with altered expression in 13 months vs 13 months+CAD121 (**B**) SAMP8 mice. Gene overlap for the KEGG pathways oxidative phosphorylation (OP), Parkinson's disease (PD), Huntington's disease (HD) and Alzheimer's disease (AD) in 13 months vs 13 months+CMS121

*Figure 3 continued on next page*

*Figure 3 continued*

SAMP8 mice is shown. n = 5–6/group. (**C**) Heatmap of the fold gene expression for 9 months/13 months, 13 months/13 months+CMS121 and 13 months/13 months+J147, regarding the mitochondrial genes present in the pathways OP, PD, HD and AD. The log2 fold-change (FC) is plotted in red–blue color scale with red indicating up-regulation and blue indicating down-regulation. ETC: Electron transport chain complex; CI = Complex I; CIII = Complex III; CIV = Complex IV; CV = Complex V. (**D**) Heatmap of the fold gene expression for 9 months/13 months, 13 months/13 months+CMS121 and 13 months/13 months+J147, regarding all genes from the mitochondrial DNA with detectable expression levels. Color key = log2 FC. (**E**) Cluster two from the clustering heatmap (K-means, Euclidean distance on normalized log2(FPKM+5) for 20077 genes) of the whole hippocampal transcriptome of 9 months, 13 months, 13 months+CMS121 and 13 months+J147 SAMP8 mice (For full heatmap see *Figure 3—figure supplement 1*). (**F**) Top KEGG pathways associated with cluster 2. (**G**) Transcriptomic drift of genes that encode for all proteins known to be associated with the mitochondria (Mitocarta). Values are expressed as box-and-whisker plots. Brown-Forsythe test. n = 5–6/group. (**H**) Mitochondrial DNA copy number in the hippocampus (normalized to 9 months). Data are mean ± SD. One-way ANOVA followed by Tukey-Kramer post-hoc test (n = 5/group).

The online version of this article includes the following figure supplement(s) for figure 3:

**Figure supplement 1.** Additional analysis of transcriptomic data.

aging and CMS121 (overlap p-value<0.01, Fisher's Exact Test). In addition, the set of DE genes in the OP pathway largely overlapped with those from the Parkinson's disease (PD), AD and Huntington's disease (HD) pathways (*Figure 3B*). These genes included nuclear genes that encode for subunits of complexes I, III and IV of the electron transport chain (ETC) and ATP synthase (complex V) in the mitochondria (*Figure 3C*). Surprisingly, the expression of all these genes went up between 9 and 13 months in the SAMP8 mice and this was prevented by CMS121 and J147 (*Figure 3C*). We also observed the same effect with genes encoded by the mitochondrial DNA, which code for proteins necessary for oxidative phosphorylation as well as the tRNA and rRNA necessary for translation in the mitochondria (*Figure 3D*).

A similar conclusion was reached when we used an alternative approach by clustering the whole transcriptome according to gene expression patterns. We found one large cluster (cluster 2) that showed increased changes in gene expression with aging that were prevented by the compounds (*Figure 3E,F*), and whose top KEGG pathways were the same as the ones identified in *Figure 3A and B*, including oxidative phosphorylation, Parkinson's disease, Huntington's disease and Alzheimer's disease. Finally, the transcriptomic drift analysis of the Mitocarta genes shows that both compounds are effective in preventing the drift of transcripts associated with mitochondrial proteins (*Figure 3G*).

The analysis of all genes that encode proteins associated with the mitochondria shows both up- and down-regulated changes in expression during aging (*Figure 3—figure supplement 1A,B*). In order to understand whether the changes in ETC gene expression were related to alterations in the number of mitochondria, we measured mitochondrial DNA copy number. We found that mitochondrial DNA copy number went down with age in SAMP8 mice and that only J147 was able to significantly prevent that decrease (*Figure 3H*). These data show that changes in mitochondrial gene expression correlate with the aging SAMP8 phenotype and are partly prevented by CMS121 and J147.

## CMS121 and J147 preserve key brain mitochondrial metabolites that are altered during aging

We next tested the hypothesis that the coordinated responses at the gene expression level with aging are associated with dysregulation at the metabolic level. To do this, we carried out large-scale, untargeted metabolic profiling of brain cortical tissue from 9 months, 13 months and 13 months+compound-treated SAMP8 mice. This study not only assessed the levels of metabolites that directly participate in brain mitochondrial metabolic pathways, such as the tricarboxylic acid (TCA) cycle and fatty acid synthesis/oxidation, but also metabolites that are involved in many other pathways. A total of 496 small molecules were measured. 205 metabolites were significantly altered between 9 and 13 months in the SAMP8 mice and the alterations in some of these metabolites were prevented by CMS121 and J147 (*Figure 4C*). Global suppression of age-related metabolomic changes by the compounds was not as robust as with the transcriptome. This observation was confirmed by the PCA and metabolic drift analyses (*Figure 4A,B*).

We also measured the levels of metabolites in the plasma of the same animals. Both compounds showed a partial suppression of the changes that took place with aging (*Figure 4E,F* and

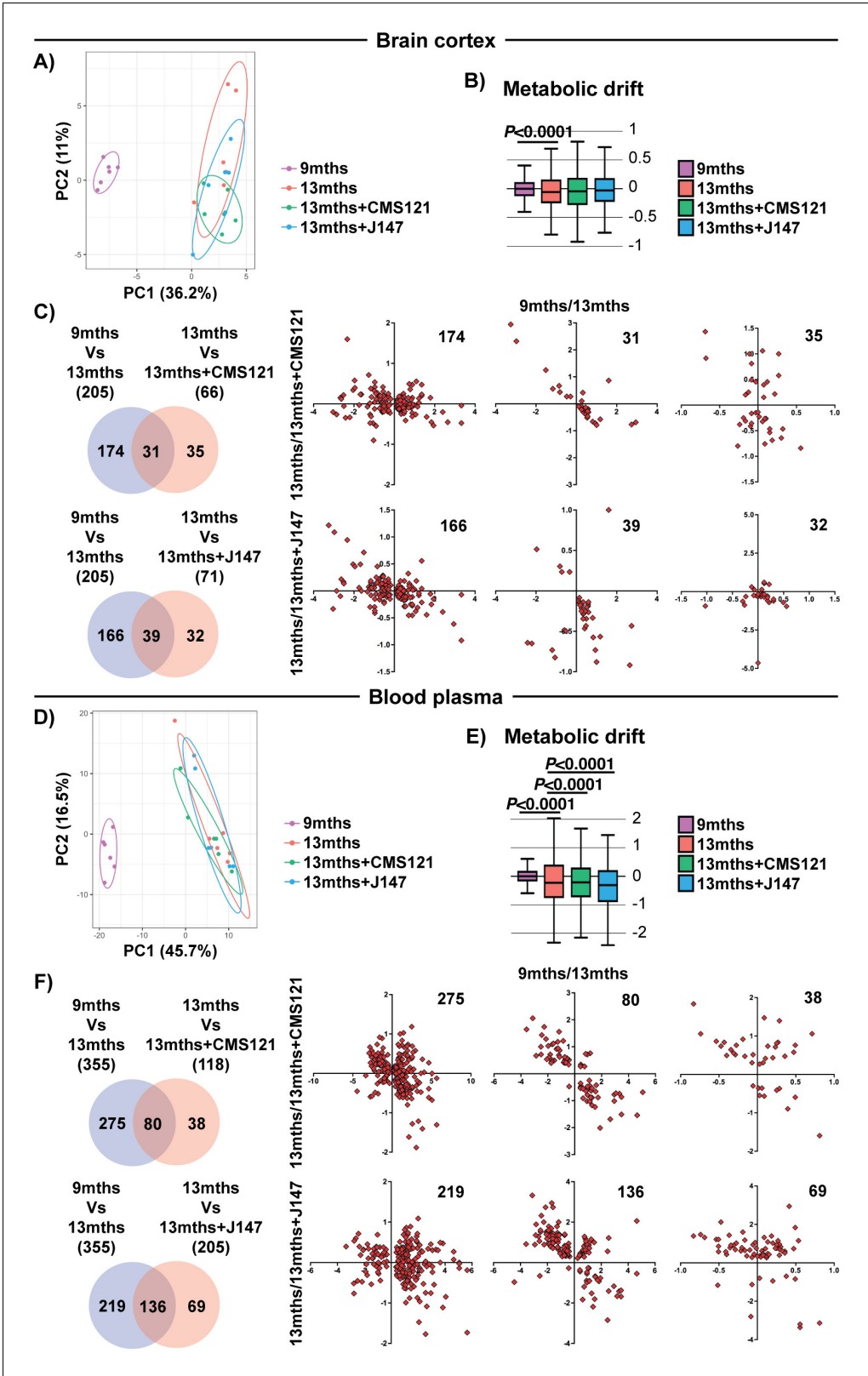

**Figure 4.** Global metabolic analysis of brain cortex and blood plasma. PCA of the 496 and 604 metabolites measured in the brain cortex (**A**) and blood plasma (**D**), respectively, of 9 months, 13 months, 13 months+CMS121 and 13 months+J147 SAMP8 mice. (n = 6/group). Drift analysis of cortex (**B**) and plasma (**E**) metabolites. Values are expressed as box-and-whisker plots. Brown-Forsythe test. n = 6/group. Venn diagrams illustrating shared and
*Figure 4 continued on next page*

*Figure 4 continued*

uniquely affected cortex (**C**) and plasma (**F**) metabolites between the 9 months vs 13 months old mice and 13 months vs each of the 13 months+compounds. n = 6/group. Correlation of metabolite levels for each group of genes is represented on the right side of the panel (units are -log(fold change)).

*Supplementary file 3*). Although not as clear in the PCA (*Figure 4D*), the metabolic drift analysis showed a significant suppression of plasma metabolic drift by CMS121 and J147 (*Figure 4E*). This is also evident in the correlation graphs for the metabolites in common between the comparisons of 9 months Vs 13 months and 13 months Vs 13 months+compounds (*Figure 4F*).

A detailed investigation of individual metabolites measured in the brain that participate in the TCA cycle revealed profound changes with aging that were partially prevented by both CMS121 and J147 (*Figure 5A*). In fact, increases of 9.51, 7.85, 6.19 and 5.73-fold between 9 and 13 months old SAMP8 mice in the levels of α-ketoglutarate, succinate, citrate and aconitate, respectively, accounted for the top largest fold changes seen with aging, while there was a 2-fold decrease in acetyl-CoA (*Figure 5A* and *Supplementary file 2*). CMS121 had the largest effect on the metabolites, significantly restoring the levels of acetyl-CoA, α-ketoglutarate and succinate to 9 months old levels (*Figure 5A*). J147 also preserved the levels of acetyl-CoA and, although not statistically significant, it reduced the increases in α-ketoglutarate and succinate seen with aging (*Figure 5A*).

Pathway enrichment analysis with all of the brain metabolites that were significantly altered with aging confirmed that the TCA cycle was among the top pathways preferentially affected (*Figure 5B*). In addition, a number of pathways that either interact with or directly take place in the mitochondria were also detected (in red), including oxidative phosphorylation, urea cycle, glutamate metabolism, pantothenate/CoA and glycolysis (*Figure 5B*).

We then carried out pathway enrichment analysis with the brain metabolites that were altered by the compounds in relation to the 13 months old mice and found that both compounds shared a common top pathway related to the metabolism of acyl carnitines (*Figure 5C and D*). The levels of the different acyl carnitines did not significantly change between 9 and 13 months, but many were increased by CMS121 and J147 in the old animals (*Figure 5E*), suggesting a shared mechanism of action for the compounds. Acyl carnitines are intermediary metabolites of fatty acid β-oxidation involved in the transport of the acyl moiety into the mitochondria where it is oxidized to produce acetyl-CoA. The levels of carnitine, which is necessary for the transport of the acyl groups across the mitochondrial inner membrane, were also significantly decreased in 13 months old mice and rescued by the compounds (*Figure 5E*).

Although no pathway analysis with the metabolic data from the plasma was pursued due to the heterogeneous nature of plasma metabolites derived from diverse tissues and organs, the data demonstrate that the compounds also had a systemic effect on some parameters of aging.

Specifically, these data show that the maintenance of mitochondrial gene expression in the brain by CMS121 and J147 is associated with a preservation of key mitochondrial metabolic intermediates. Moreover, the build-up in acyl carnitines with both compounds is suggestive of a mitochondrial target underlying the action of the compounds.

## CMS121 and J147 enhance acetyl-CoA levels and histone acetylation in neurons

To further delineate the overall changes in metabolism shared by both compounds, network analysis was carried out by GAM integrating both the transcriptomic and the metabolomic data. Analysis of the networks generated for each compound showed that acetyl-CoA is a central metabolite shared by CMS121 and J147 (*Figure 5—figure supplement 1A,B*). To rule out the possibility of capturing acetyl-CoA in the network just by chance, random sampling of the transcriptomic and the metabolomics data was performed followed by network analysis, and confirmed the central role of acetyl-CoA in the experiments (random sampling p=0.01). Importantly, the decrease in the levels of acetyl-CoA in the brains of 13 months old SAMP8 mice was prevented by CMS121 and J147 (*Figure 5A*).

To provide further support for the central role of acetyl-CoA in the effects of the compounds, network analysis was performed on transcriptomic data from cultured, differentiated rat cortical neurons treated with CMS121 and J147. This analysis also identified acetyl-CoA as a central metabolite regulated by both compounds (*Figure 5—figure supplement 1C,D*; random sampling, p=0.01).

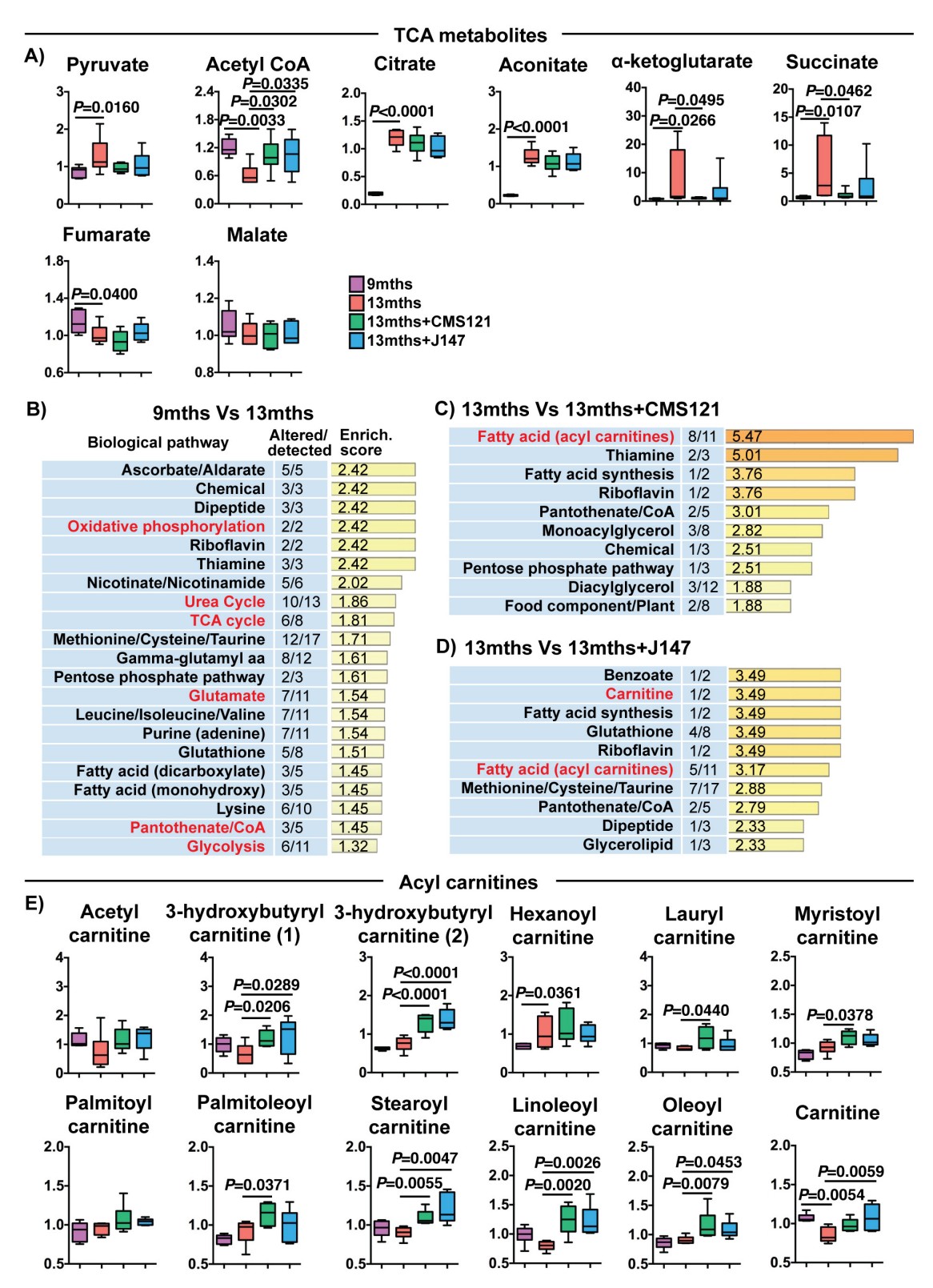

**Figure 5.** CMS121 and J147 modulate key metabolites associated with mitochondria in the brain of SAMP8 mice. (**A**) Levels of TCA metabolites detected in the cortex of 9 months, 13 months, 13 months+CMS121 and 13 months+J147 SAMP8 mice. Values are expressed as box-and-whisker plots. One-way ANOVA followed by Tukey-Kramer post-hoc test (n = 6/group). Pathway enrichment analysis with cortex metabolites found altered in 9 months vs 13 months (**B**), 13 months vs 13 months+CMS121 (**C**), and 13 months vs 13 months+J147 (**D**). Enrichment scores were calculated as described

*Figure 5 continued on next page*

*Figure 5 continued*

in the Supplemental Materials and methods. Number of metabolites altered in a biological pathway and the total number of metabolites measured in that pathway (Altered/detected) are shown as a complement to the enrichment score. (E) Levels of all acyl carnitine metabolites measured in the cortex of 9 months, 13 months, 13 months+CMS121 and 13 months+J147 SAMP8 mice. Values are expressed as box-and-whisker plots. One-way ANOVA followed by Tukey-Kramer post-hoc test (n = 6/group).

The online version of this article includes the following figure supplement(s) for figure 5:

**Figure supplement 1.** Network analysis.

Measurement of acetyl-CoA in these cells confirmed that both compounds induced a significant increase (*Figure 6A*).

Acetyl-CoA is used by histone acetyltransferases to acetylate histones and thereby regulate memory by restructuring chromatin in the brain (*Mews et al., 2017*). In primary cortical neurons, CMS121 and J147 significantly increased acetylation of histone 3 at lysine 9 (H3K9) (*Figure 6B*), an epitope that has been implicated in memory formation (*Fischer et al., 2007*; *Mews et al., 2017*). Importantly, acetylation of H3K9 was reduced in the cortex of 13 months old SAMP8 mice relative to 9 months old mice and both CMS121 and J147 significantly prevented this loss (*Figure 6C*). We obtained similar results in the APPswe/PSEN1dE9 transgenic mouse model of AD in which old symptomatic mice were fed CMS121 and J147 that resulted in improved cognition and reduced pathology (*Prior et al., 2013* and data submitted for publication). In both cases, acetylation of H3K9 was increased by the compounds (*Figure 6—figure supplement 1D–G*). These data identify acetyl-CoA as a central metabolite that is decreased in the aging brain and increased by CMS121 and J147, and this increase is associated with acetylation of histone H3 at a site required for memory enhancement.

## Increases in acetyl-CoA levels are neuroprotective and are associated with inhibition of ACC1 by CMS121 and J147

We have recently demonstrated that the mechanism of action of J147 is associated with the activation of AMPK (*Goldberg et al., 2018*). One of the downstream effects of AMPK activation is the inhibition by phosphorylation at serine 79 of acetyl-CoA carboxylase 1 (ACC1), an enzyme responsible for the conversion of acetyl-CoA into malonyl-CoA, which is then utilized to synthesize fatty acids. The flux of acetyl-CoA through ACC1 to malonyl-CoA is a major mechanism of acetyl-CoA catabolism and lipid biosynthesis. Inhibition of ACC1 can thus potentially lead to a build up in acetyl-CoA. Therefore, we determined the effects of the compounds on ACC1 phosphorylation and acetyl CoA levels in both primary neurons and the mouse hippocampal neuronal cell line HT22, the primary phenotypic screening model used in our laboratory to identify CMS121 and J147 (*Chen et al., 2011*; *Chiruta et al., 2012*) (*Figure 6D*). This assay relies on the lethal induction of mitochondrial ROS in the HT22 cells (*Prior et al., 2014*), a programmed cell death pathway called oxytosis with physiological features similar to those implicated in the nerve cell damage seen in aging and AD (*Currais and Maher, 2013*; *Tan et al., 2001*). Oxytosis can be triggered by glutamate which inhibits cystine uptake via system $X_c^-$, leading to depletion of intracellular glutathione (GSH), production of ROS and cell death (*Tan et al., 2001*).

We first determined that the compounds can increase acetyl-CoA in these cells (*Figure 6E*). In HT22 cells as well as primary neurons, both CMS121 and J147 increased the phosphorylation of ACC1 at serine 79 (*Figure 6F,G*; Western blots shown in *Figure 6—figure supplement 1*). Importantly, both CMS121 and J147 also increased phospho-ACC1 levels in the brains of the SAMP8 mice (*Figure 6H*). ACC1 phosphorylation was also elevated with aging, and this observation could be associated with a compensatory mechanism. In addition, inhibition of ACC1 by the compounds was accompanied by a decrease in free fatty acids in primary neurons (*Figure 6I*) as well as in the brains (*Figure 6J*) and plasma (*Figure 6K*) of old SAMP8 mice. Thus, inhibition of ACC1 can at least partially explain the increases in acetyl-CoA levels seen with both compounds (*Figure 6A and E*).

To test this hypothesis directly, ACC1 activity was blocked with the competitive inhibitor 5-(tetradecyloxy)−2-furoic acid (TOFA) or ACC1 expression was reduced with a specific siRNA. Both of these approaches significantly increased the levels of acetyl-CoA in the HT22 cells (*Figure 6L and N*). If acetyl-CoA is a key metabolite that contributes to the biological activity of CMS121 and J147, then inhibiting ACC1 activity or reducing its levels should phenocopy the compounds'

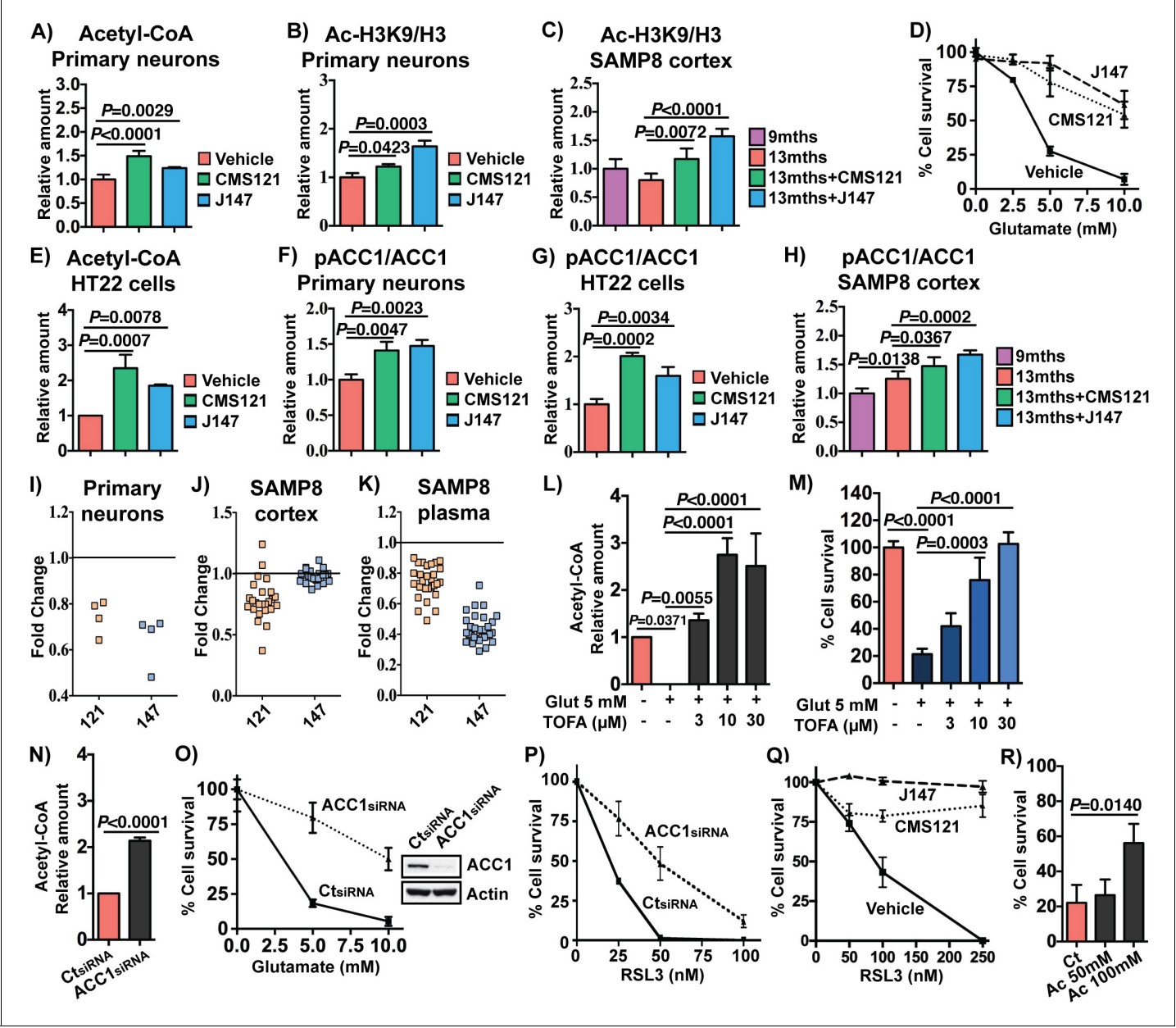

**Figure 6.** CMS121 and J147 increase acetyl-CoA levels associated with inhibition of ACC1. Increases in acetyl-CoA levels are protective and associated with inhibition of ACC1 by CMS121 and J147. (**A**) Total levels of acetyl-CoA in rat primary neurons (n = 5/group), after treatment with 1 μM of each compound for 24 hr. Acetylation of H3 at Lys 9 (Ac-H3K9) was assessed by Western blotting and normalized to levels of total H3 in rat primary neurons (n = 3/group) (**B**) and mouse cortical tissue (n = 5/group) (**C**). (**D**) Protection during oxytosis in HT22 cells with CMS121 (500 nM) and J147 (50 nM) (n = 3/group). (**E**) Total levels of acetyl-CoA in HT22 cells (n = 3/group). Phosphorylation of ACC1 was assessed by Western blotting and normalized to levels of respective total protein in rat primary neurons (n = 3/group) (**F**), HT22 cells (n = 3/group) (**G**) and SAMP8 mouse cortical tissue (n = 5/group) (**H**). Scans of the blots are shown in *Figure 6—figure supplement 1*. (**I**) Fold change in the levels of different polyunsaturated fatty acids in primary neurons treated with 1 μM of each compound for 24 hr. Fold change in the levels of multiple long-chain fatty acids including major polyunsaturated fatty acids in the brains (**J**) and plasma (**K**) of old SAMP8 treated with CMS121 and J147, relative to old SAMP8. (**L**) Total levels of acetyl-CoA in HT22 cells treated with 5 mM glutamate alone or in the presence of 3, 10 or 30 μM TOFA, determined 8 hr after treatment (n = 3/group). (**M**) Protection during oxytosis in HT22 cells after inhibition of ACC1 with 3, 10 or 30 μM TOFA (n = 3/group). (**N**) Total levels of acetyl-CoA in HT22 cells after knock down with ACC1 siRNA, measured 48 hr after knock down (n = 3/group). Protection during oxytosis (**O**) and ferroptosis (**P**) in HT22 cells after knock down with ACC1 siRNA (n = 3/group). (**Q**) Protection during ferroptosis in HT22 cells with CMS121 (250 nM) and J147 (100 nM) (n = 3/group). (**R**) Protection during oxytosis in HT22 cells treated with sodium acetate (Ac), with survival normalized to control treatment with sodium chloride. All data are mean ± SD. One-way ANOVA followed by Tukey-Kramer post-hoc test.

The online version of this article includes the following figure supplement(s) for figure 6:

*Figure 6 continued on next page*

*Figure 6 continued*

**Figure supplement 1.** Phosphorylation of ACC1 and acetylation of H3 in primary neuronal cultures, HT22 cells and brain cortex of SAMP8 mice by CMS121 and J147.
**Figure supplement 2.** Inhibition of ACC1 by CMS121 and J147 is dependent on AMPK.

neuroprotective effects. *Figure 6M and O* show that both approaches were neuroprotective in HT22 cells exposed to glutamate. Similarly, both CMS121 and J147 as well as knocking down ACC1 expression protected HT22 cells from ferroptosis induced by RSL3 (*Figure 6P and Q*), a non-apoptotic form of regulated cell death that is similar to oxytosis, including mitochondrial dysfunction (*Lewerenz et al., 2018*; *Tan et al., 2001*). In addition, consistent with the hypothesis that increases in acetyl CoA levels are protective, directly increasing acetyl-CoA levels by supplying cells with acetate, a cellular precursor of acetyl-CoA, was also protective (*Figure 6R*).

Finally, similar to what we found previously with J147 (*Goldberg et al., 2018*), the inhibition of ACC1 by CMS121 was associated with AMPK activation (*Figure 6—figure supplement 2*). In addition, using cells that do not express AMPK, we found that the increases in acetyl-CoA levels and neuroprotection as a consequence of ACC1 inhibition are dependent on AMPK (*Figure 6—figure supplement 2A–C*).

Together, these data show that the preservation of brain mitochondrial gene expression and metabolism in the aging brain by both CMS121 and J147 is mediated by a shared mechanism involving the inhibition of ACC1 and increased levels of the central mitochondrial metabolite acetyl-CoA.

## Discussion

Despite the observation that old age is the greatest risk factor for AD, few clinical approaches have focused on treating AD and other dementias by targeting detrimental age-related processes that are relevant to disease onset and development. In part, this is because we still lack a deep understanding of the aging brain. As such, our laboratory has developed a novel drug discovery program based upon cell-based screening assays that mimic multiple aspects of old age-associated neurodegeneration and AD pathology (*Prior et al., 2014*) and has used a rapidly aging mouse model to study drug candidate effects in the context of aging (*Currais et al., 2015b*). This approach has led to the identification of CMS121 and J147, two potent neuroprotective small molecules. Because these compounds have different structures, but are effective in the same in vitro models of toxicities associated with the aging brain, we asked if they could be used to identify metabolic changes associated with age-related cognitive dysfunction, leading to the identification of novel therapeutic targets.

CMS121 and J147 were able to reduce age-related cognitive dysfunction, even when administered to SAMP8 mice at a late stage of their lives. In addition, CMS121 and J147 share a common mechanism of action that is associated with the maintenance of mitochondrial health in terms of transcriptional stability and metabolism. Mitochondrial dysfunction is one of the hallmarks of aging (*López-Otín et al., 2013*), and an acceleration in this dysfunction may be responsible for the onset of AD pathology (*Swerdlow and Khan, 2004*; *Yin et al., 2016*). Therefore, mitochondria represent an important therapeutic target for the treatment of AD (*Caldwell et al., 2015*; *Onyango et al., 2016*).

Consistent with these observations, CMS121 and J147 reduced several molecular parameters of brain aging. PCA and drift analysis of the transcriptome showed that the two compounds significantly prevented changes in gene expression associated with aging. The level of suppression by CMS121 was particularly remarkable as evidenced in the correlation plots. Although not as strong, J147 also significantly prevented transcriptomic aging, which is in accordance with what we found in the SAMP8 mice treated from a young age (3 months old) with this drug candidate (*Currais et al., 2015b*).

The drift suppression was not as strong at the level of the brain metabolites as it was with gene expression. Instead, numerous metabolites that did not change with age were directly altered by the compounds. The effects of CMS121 and J147 on certain metabolic pathways and individual metabolites were very specific, namely the increase in acyl carnitines. Since acyl carnitines are direct substrates for fatty acid oxidation in mitochondria and a source of acetyl-CoA, they further support the

conclusion that the compounds have a direct effect on mitochondrial metabolism in the brain. CMS121 and J147 also suppressed the metabolic drift in the plasma, indicating that the compounds may prevent the aging process at the systemic level as well.

Pathway analysis of the transcriptome identified genes that code for proteins that are part of the ETC and ATP synthase as being preferentially maintained at the younger level by the compounds. Mitochondria-related genes were also the top pathways affected by aging, indicating that mitochondrial dysfunction is part of the SAMP8 aging process. Because a decrease in mitochondrial DNA copy number was found with aging, the increased expression of gene transcripts associated with the ETC with aging is likely a compensatory response to a lack of adequate mitochondrial metabolism. The conclusion is supported by the observation that deficits in the respiratory chain and mitochondrial bioenergetics increase as a function of age (*Caldwell et al., 2015*; *López-Otín et al., 2013*), and by our data showing significant alterations in a number of TCA cycle metabolites in 13 months old SAMP8 mice.

Acetyl-CoA is a key metabolite that bridges glycolysis, fatty acid β-oxidation, fatty acid synthesis and the TCA cycle in the mitochondria. Its metabolism in the TCA cycle leads to the production of metabolites that are used in other metabolic pathways essential for the proper functioning of the cell, including the neurotransmitter acetylcholine which is reduced in forebrain neurons at early stages of the disease (*Mufson et al., 2008*). Acetyl-CoA is also used by histone acetyltransferases to acetylate histones and thus regulate memory formation (*Mews et al., 2017*). In addition, the TCA cycle generates reducing potential that is used by the ETC. Therefore, our data have substantial implications for the therapeutic use of CMS121 and J147 in AD, given that deficits in mitochondrial metabolism are thought to play a key role in the cognitive dysfunction of AD patients.

Although acetyl-CoA can be derived from multiple sources, our data argue that the elevation of acetyl-CoA by the compounds is due to the inhibition of ACC1 via its phosphorylation by AMPK. This conclusion is based on the following observations: 1) both compounds cause ACC1 phosphorylation in vivo, primary neurons, HT22 cells and fibroblasts, and increase acetyl-CoA levels in all four models; 2) knockdown of ACC1 increases acetyl-CoA levels and provides neuroprotection; 3) chemical inhibition of ACC1 also increases acetyl-CoA levels and provides neuroprotection; and 4) absence of AMPK prevents the inhibition of ACC1, increases in acetyl-CoA levels and reduces the protection against oxytosis.

One of the few clinical approaches targeting mitochondrial metabolism that has been tested in the context of AD is the use of ketogenic diets. In animals, ketone bodies are mainly produced in the liver from fatty acid β-oxidation through acetyl-CoA and are then transported via blood to other tissues, such as the brain, where they are once again metabolized to generate acetyl-CoA. Administration of a variety of ketogenic diets has been shown to have some positive effects in both AD transgenic mice as well as human AD patients (reviewed in *Lange et al., 2017*). Because CMS121 and J147 increase acetyl-CoA levels in the brains of animals and in vitro, our data suggest a therapeutic alternative to the ketogenic diet that leads to an improvement in mitochondrial health as well as an improvement in other parameters of aging.

Acetyl-CoA, along with several mitochondrial metabolites, has been directly implicated in the regulation of gene expression via the modification of chromatin (reviewed in *Kinnaird et al., 2016*). Histone acetylation requires acetyl-CoA that is synthesized in the cytoplasm and nucleus from acetate, citrate or pyruvate. These metabolites are intermediates of the TCA cycle and were all found to be altered with aging in SAMP8 mice. Other TCA metabolites, such as fumarate and succinate have been identified as inhibitors of Jumonji-C (JMJC) domain-containing histone demethylases (JHDMs) and TET demethylases, which rely on the TCA metabolite α-ketoglutarate as a co-substrate. The levels of all these metabolites were also altered in the brains of 13 months old SAMP8 mice. CMS121 and J147 decreased the levels of succinate and α-ketoglutarate. Therefore, it is not unexpected that alterations in mitochondrial metabolism in the 13 months old SAMP8 mice would be followed by changes at the level of gene expression.

It should be noted that, due to limitations in the amount of material, the transcriptomic analysis was carried out with brain hippocampus and that the metabolomic study was performed with brain cortex. Despite being from two different brain regions, the interpretation of the global effects of the compounds on these parameters with aging still hold valid. This is because network analysis using GAM identified acetyl-CoA as a central metabolite regulated by CMS121 and J147 in both cortex and hippocampus, and this finding was validated in primary neurons and HT22 cells. In addition,

both transcriptomics and metabolomics also identified mitochondrial dysfunction as a feature of aging in SAMP8 mice in both tissues.

The link between brain acetyl-CoA metabolism and behavioral improvement caused by CMS121 and J147 may be the increased acetylation of H3K9, known to regulate the expression of genes involved in memory formation (*Mews et al., 2017*). H3K9 acetylation was reduced in the brains of old SAMP8 mice as well as in the brains of APPswe/PSEN1dE9 transgenic AD mice. Both CMS121 and J147 increased acetylation at this site in the mouse brains as well as in primary cortical neurons. The ability of these compounds to prevent a large number of transcriptomic changes with aging might thus be a consequence of their direct effects on acetyl-CoA-mediated histone modifications.

The reason that the compounds share a common effect on metabolism is likely related to the fact that the same cell-based screening platform was used to identify both. In fact, some of these assays are characterized by mitochondrial dysfunction, such as the glutamate-directed oxytosis assay in HT22 cells (*Prior et al., 2014*). We show that, despite engaging distinct initial molecular targets in the cell, the compounds activate mechanisms that overlap at the level of mitochondrial metabolism, leading to the same phenotypic outcome.

In conclusion, when administered at advanced stages of aging and cognitive dysfunction, two structurally distinct AD drug candidates preserve brain mitochondrial homeostasis during this period of life, making mice appear younger at the level of cognition, transcription and metabolism. The analysis of molecular pathways based upon data overlapping the action of CMS121 and J147 and respective mechanistic validation identified a unique neuroprotective mechanism that maintains acetyl-CoA levels in cell culture and animal models of aging and AD (*Figure 7*). Because mitochondrial dysfunction is a hallmark of both old age and AD, it follows that other therapies that target acetyl-CoA in the brain may preserve mitochondrial homeostasis and prevent the metabolic deficits in AD patients that occur with aging.

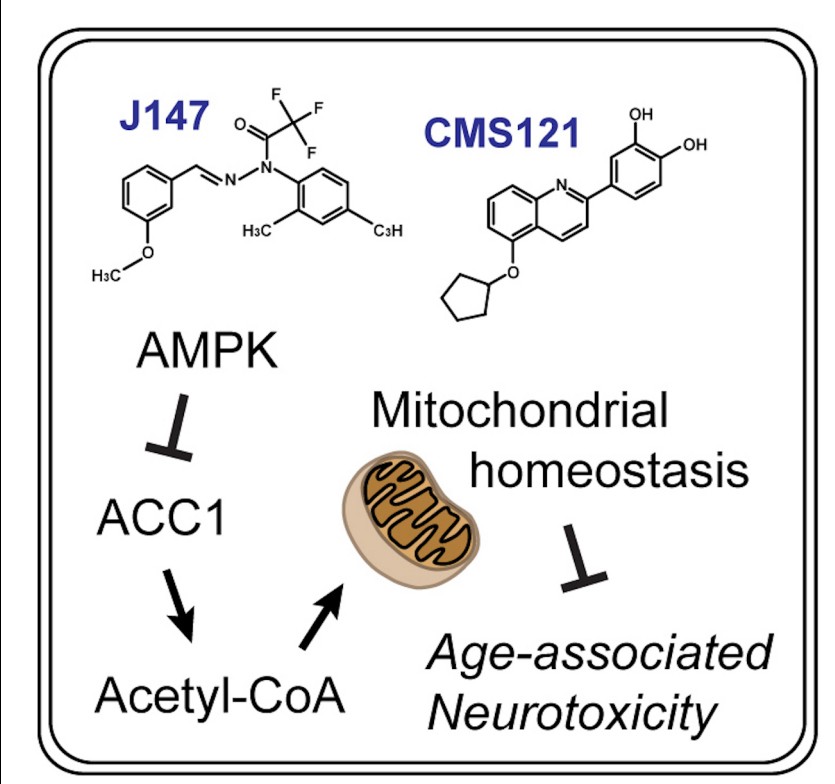

**Figure 7.** Summary diagram. CMS121 and J147 preserve brain mitochondrial homeostasis with aging via a neuroprotective mechanism that regulates acetyl-CoA metabolism.

# Materials and methods

## Study design

Twenty-three 9 months old female SAMP8 mice were fed with vehicle diet (LabDiet 5015, TestDiet, Richmond, IN), and twenty-two and twenty-two 9 months old female SAMP8 mice were fed with CMS121 and J147 diets, respectively (LabDiet 5015 + 400ppm CMS121, LabDiet 5015 + 200ppm J147, TestDiet). Diet treatment lasted for four months until mice reached 13 months of age. At 9 months of age, SAMP8 mice already present a strong phenotype (*Currais et al., 2015b*; *Currais et al., 2012*; *Takeda, 2009*). The dose of J147 used was 200 ppm (~10 mg/kg/day), which previously proved effective in mouse models (*Currais et al., 2015b*; *Prior et al., 2013*; *Prior et al., 2016*). For CMS121 we used 400 ppm (~20 mg/kg/day), which was chosen based on its greater efficacy than its parent, fisetin, in in vitro assays and positive results with 500 ppm fisetin in the SAMP8 mice (*Currais et al., 2018*). Eleven 9 months old female SAMP8 mice were used as the baseline control group.

The effect of the compounds was assessed in older SAMP8 mice after the four months of treatment and any age-related changes are defined by comparison to 9 months old SAMP8 mice. All mice were randomly assigned to experimental groups. The number of mice per group was determined based on previous experiments (*Currais et al., 2015b*; *Currais et al., 2012*) and was sufficient to attain statistical power. Six SAMP8 mice fed with control diet, four SAMP8 mice fed with CMS121 diet and four SAMP8 mice fed with J147 diet died throughout the course of this study. Behavioral testing was carried out one month prior to sacrifice and collection of biological material. Data were analyzed by blinded researchers when appropriate.

## SAMP8 mice

The SAMP8 line, a naturally occurring mouse line that was developed based on its phenotype of accelerated aging, was acquired from Harlan Laboratories (U.K.). Mouse body weights were measured regularly and no significant differences were found between the groups (*Figure 2—figure supplement 2*). All experiments were performed in accordance with the US Public Health Service Guide for Care and Use of Laboratory Animals and protocols approved by the IACUC at the Salk Institute.

## Tissue preparation

Mice were anesthetized and their blood collected by cardiac puncture. After perfusing with PBS, their brains were removed and dissected to collect cortex and hippocampus. Tissue was prepared for Western blotting, RNA extraction and metabolomic analysis.

## Large-scale metabolome analysis

Metabolite measurement and analysis were conducted at Metabolon as previously described (*Currais et al., 2015b*).

## Measurement of acetyl-CoA

Acetyl-CoA levels were determined in protein-free lysates of primary neurons or HT22 cells treated as indicated in the figure legends using a kit from Sigma (MAK039) according to the manufacturer's instructions. The levels were normalized to the protein in the solubilized pellet using the BCA assay.

## Supplemental materials and methods

### Cell lines

Mouse hippocampal HT22 cells, generated in our laboratory, were propagated as previously described (*Davis and Maher, 1994*). AMPK knockout (K.O.) mouse embryonic fibroblasts were a generous gift from Ruben Shaw (Salk Institute). The lack of AMPK expression can be verified by Western blotting (*Figure 6—figure supplement 2*). To prevent cell misidentification, large batches of each cell line were frozen that are regularly thawed to avoid using the wrong cell line. Cell lines are routinely tested for mycoplasma and no contamination has been found.

## Primary neurons

Primary cortical neurons were prepared from day 17 rat embryos and used at 7 days in vitro (7 DIV) (*Soucek et al., 2003*).

## APPswe/PSEN1dE9 transgenic AD mice

Samples from line 85 APPswe/PSEN1dE9 transgenic AD mouse mice and respective wild type (Wt) mice treated with J147 in a reversal experimental paradigm were obtained from our previous study (*Prior et al., 2013*). In another independent study, 10 months old male Wt and APPswe/PSEN1dE9 mice were fed with vehicle diet (LabDiet 5015, TestDiet, Richmond, IN) or CMS121 diet (LabDiet 5015 + 400ppm CMS121) for three months and brain cortical tissue collected.

## Behavioral assays

Elevated plus maze - The maze consisted of four arms (two open without walls and two enclosed by 15.25 cm high walls) 30 cm long and 5 cm wide in the shape of a plus. A video-tracking system (Noldus EthoVision) was used to automatically collect behavioral data. The software was installed on a PC computer with a digital video camera mounted overhead on the ceiling, which automatically detected and recorded when mice entered the open or closed arms of the maze and the time spent in each. Mice were habituated to the room 24 hr before testing and habituated to the maze for 1 min before testing by placing them in the center of the maze and blocking entry to the arms. Mice were then tested for a 5 min period and their behavior recorded. Disinhibition was measured by comparing the time spent on the open arms to time spent on the closed arms.

Barnes maze - The maze consisted of a flat circular surface (36' diameter) with 20 equally spaced holes (2' diameter) along the outer edge. One of the holes led to a dark hide box while the other 19 led to false boxes that were too small to be entered. The latency to enter the hide box was recorded. The test was conducted in three phases. Phase 1 (Training): A hide box was placed under one of the holes. Animals were placed into an opaque cylinder in the center of the maze for 30 s to promote spatial disorientation at the start of the test. After 30 s, the cylinder was removed and the animal explored the maze until it found and entered the hide box. The number of incorrect entries was scored. If the mouse failed to enter the box within 3 min, it was gently led into the box. The animal remained in the box for an additional 20 s before it was removed from the boxed and gently placed into the home cage. Training is repeated three times a day for four days. The location of the hide box remained the same during every trial but it was shifted between subjects to reduce the potential for unintended intra-maze cues. Phase 2 (Retention): This phase measures retention of spatial memory following a delay. After a two day break from training, each animal was re-tested for a one day, three-trial session using the same hide box location as before. Phase 3 (Reversal): This phase examines memory reversal. On the day following the retention phase, a new hide box location was established 180 degrees to the original location. The same method as before was used and trials were repeated three times a day over two consecutive days.

## Western blotting

Western blots were carried out as described previously (*Currais et al., 2015b*). The primary antibodies used were: HRP-conjugated rabbit anti-actin (#5125, 1/20,000), acetyl-histone H3 (Lys9) (#9649, 1/100000), phospho-ACC1 (#3661, 1/2000), total ACC1 (#4190, 1/1000), phospho-AMPK (#2535, 1/1000) and total AMPK (#2793, 1/1000), from Cell Signaling Technology; histone H3 (#ab24834, 1/100000) from Abcam. Horseradish peroxidase-conjugated secondary antibodies (goat anti-rabbit, goat anti-mouse or rabbit anti-goat (BioRad) diluted 1/5000) were used.

## Whole transcriptome analysis

RNA was isolated from the hippocampus of SAMP8 mice or primary neurons using the RNeasy Plus Universal mini kit (Qiagen). RNA-Seq libraries were prepared using the Illumina TruSeq Stranded mRNA Sample Prep Kit according to the manufacturers instructions. Briefly, poly-A RNA was selected using poly dT-beads. mRNA was then fragmented and reverse transcribed. cDNA was end-repaired, adenylated and ligated with Illumina adapters with indexes. Adapter-ligated cDNA was then amplified. Libraries were pooled and sequenced single-end 50 base-pair (bp) on the Illumina HiSeq 2500 platform. Sequencing reads were mapped to the mm10 mouse genome or rn6 rat

genome using the spliced aligner STAR (2.5.1b) with default parameters (*Dobin et al., 2013*). Reference mm10 mouse and rn6 rat genome was downloaded from UCSC. Only uniquely aligned reads were considered for downstream analysis. Expression values were quantified using Homer (4.9.1, *Heinz et al., 2010*) by counting reads mapped across all gene exons of RefSeq genes and mitochondrial encoded genes. The differential expression (DE) analysis was performed by edgeR (v3.16.1, *McCarthy et al., 2012*). Briefly, genes with counts per million greater than one for at least half of the samples were normalized by the default 'TMM' method. The dispersion was estimated by the 'estimateGLMTagwiseDisp' function. Genes with a false discovery rate (FDR) < 0.05 and an absolute log2 fold-change >0.3 (about 1.23 fold-change) were identified as significantly differentially expressed. The data discussed in this publication have been deposited in NCBI's Gene Expression Omnibus (*Edgar et al., 2002*) and are accessible through GEO Series accession number GSE101112 (https://www.ncbi.nlm.nih.gov/geo/query/acc.cgi?acc=GSE101112).

## Transcriptome/metabolome drift analysis

Transcriptional drift analysis was performed as previously described (*Rangaraju et al., 2015*) with the exception that we removed expressed genes below the 20th percentile. We further normalized all samples setting the overall mean transcriptional drift to 0 to avoid differences in sample counts that affect drift variance across samples. Metabolomic drift variance was calculated by first determining the metabolomics drift (md) of each individual metabolite (md = log(old/young). The metabolomics drift variance is then determined by calculating the variance of md across the entire metabolome or subgroups of the metabolome.

## Mitochondrial DNA copy number

Mitochondrial DNA copy number was determined in the hippocampus of SAMP8 mice using the kit from Detroit R and D, Inc (MCN 3) according to the manufacturer's instructions.

## Bioinformatics and statistics

R package 'gplots' (*Warnes et al., 2016*) was used to generate the heatmaps and MSEAs. For the RNA and metabolic heatmaps, all values were mean-centered and divided by the SD of each variable (scaled Z-score). The heatmap of RNA-Seq was z-scaled log2(FPKM+5) for *Figure 3E* and *Figure 3—figure supplement 1C* and log2 fold-change for *Figure 3C and D* and *Figure 3—figure supplement 1A*. Hierarchical clustering of RNA expression was performed using Euclidean distances and the Ward2 algorithm. K-means clustering of RNA expression was performed using Euclidean distances on z-scaled log2(FPKM+5) of 20077 expressed genes (at least one sample had greater than 0 FPKM). The total number of 20077 expressed genes were used as background for the DAVID pathway analysis for k1 cluster. MSEAs were generated according to HMDB and the 'Pathway-associated metabolic sets'. Only top pathways are indicated.

PCA of the transcriptome data was performed on the log2(FPKM+5) values of the top 10% most expressed genes (cutoff: RowSums(log2(FPKM+5))>=110.41; 2458 genes selected) by the R base function 'prcomp' with data mean centered. Metabolome data was imputed to replace the missing value by the minimal value observed across all samples and transformed by the function 'glog' from the R package 'FitAR' (W. Huber, A. von Heydebreck, H. Sultmann, A. Poustka, and M. Vingron. Variance stablization applied to microarray data calibration and to quantification of differential expression. Bioinformatics, 18: S96-S10 2002) before PCA analysis. The ellipses showed the 70% confidence interval of a multivariate normal-distribution of sample groups estimated by PC1 and PC2 data.

Pathway enrichment test of the transcriptome data was analyzed by DAVID Bioinformatics Resources v6.8 (*Huang et al., 2009a* and *Huang et al., 2009b*). The total number of expressed genes of each individual pair-wise comparisons was used as background for the enrichment test.

Pathway enrichment analysis on metabolic pathways was conducted using the MetaboLync platform (Metabolon). Enrichment scores = (number of significant metabolites in pathway/total number of detected metabolites in pathway)/(total number of significant metabolites/total number of detected metabolites).

The metabolic network was reconstructed by GAM (*Sergushichev et al., 2016*) with input of the metabolome and/or the transcriptome dataset(s) and visualized by Cytoscape (*Shannon et al., 2003*).

Random Sampling: The dataset that contained the test statistics (p-value and fold-change) between 13 months+CMS121 and 13 months was used for the random sampling experiment. The Gene IDs and the Compound IDs were randomly shuffled 10 times (sampling without replacement) followed by the network analysis using the same parameters and cutoffs. The probability of observing acetyl-CoA in the resulted network was calculated as pnull. Random sampling $p=pnull^{na}*(1-pnull)^{nb}$ (na = number of experiments that identified acetyl-CoA in the network; nb = number of experiments that did not identify acetyl-CoA in the network). In this experiment, pnull = 0.1 and random sampling p=0.01. For the rat primary neuron transcriptome data, random sampling was performed using the same method with data generated from the comparison between CMS121 and control.

GraphPad Prism six was used for statistical analysis and exact *P* values are indicated (for $p<0.050$).

## Acknowledgements

We thank Joseph Chambers for help with husbandry of mice.

## Additional information

### Competing interests

David Schubert: is an unpaid advisor for Abrexa Pharmaceuticals, a company working on the development of J147 for AD therapy. The Salk Institute holds the patents for CMS121 and J147. The other authors declare that no competing interests exist.

### Funding

| Funder | Grant reference number | Author |
| --- | --- | --- |
| National Institutes of Health | R01 AG046153 | David Schubert<br>Pamela Maher |
| National Institutes of Health | RF1 AG054714 | David Schubert<br>Pamela Maher |
| Glenn Foundation for Medical Research | | Joshua Goldberg |
| National Institutes of Health | R41 AI104034 | Pamela Maher |
| Edward N. and Della L. Thome Memorial Foundation | | Pamela Maher |

The funders had no role in study design, data collection and interpretation, or the decision to submit the work for publication.

### Author contributions

Antonio Currais, Conceptualization, Data curation, Formal analysis, Supervision, Investigation, Methodology, Writing—original draft, Project administration, Writing—review and editing; Ling Huang, Data curation, Formal analysis, Validation, Investigation, Methodology, Writing—review and editing; Joshua Goldberg, Data curation, Formal analysis, Writing—review and editing; Michael Petrascheck, Data curation, Formal analysis, Methodology, Writing—review and editing; Gamze Ates, António Pinto-Duarte, Formal analysis, Investigation, Writing—review and editing; Maxim N Shokhirev, Formal analysis, Writing—review and editing; David Schubert, Conceptualization, Supervision, Writing—review and editing; Pamela Maher, Conceptualization, Formal analysis, Validation, Investigation, Writing—review and editing

Author ORCIDs
Antonio Currais (iD) https://orcid.org/0000-0003-4142-7054
Michael Petrascheck (iD) http://orcid.org/0000-0002-1010-145X
António Pinto-Duarte (iD) http://orcid.org/0000-0002-2215-7653

## Ethics

Animal experimentation: All experiments were performed in accordance with the US Public Health Service Guide for Care and Use of Laboratory Animals and protocol 12-00001 approved by the IACUC at Salk Institute.

## Decision letter and Author response
Decision letter https://doi.org/10.7554/eLife.47866.sa1
Author response https://doi.org/10.7554/eLife.47866.sa2

## Additional files

### Supplementary files

• Supplementary file 1. List of the DE genes found in 9 vs 13 months old SAMP8 mice and 13 months vs 13 months+compounds SAMP8 mice.

• Supplementary file 2. List of all 496 metabolites quantified in the brain cortex of 9 months, 13 months, 13 months+CMS121 and 13 months+J147 SAMP8 mice.Fold changes and specific P values are indicated. One-way ANOVA (n = 6/group).

• Supplementary file 3. List of all 604 metabolites quantified in the blood plasma of 9 months, 13 months, 13 months+CMS121 and 13 months+J147 SAMP8 mice.Fold changes and specific P values are indicated. One-way ANOVA (n = 6/group).

• Transparent reporting form

• Reporting standard 1. The ARRIVE Guidelines Checklist.

### Data availability

Whole transcriptomic data have been deposited in NCBI's Gene Expression Omnibus and are accessible through GEO Series accession number GSE101112.

The following dataset was generated:

| Author(s) | Year | Dataset title | Dataset URL | Database and Identifier |
|---|---|---|---|---|
| Currais A, Huang L | 2017 | Whole transcriptome analysis of brain hippocampal tissue from SAMP8 mice and rat primary neurons treated with the Alzheimer's disease drug candidates CMS121 and J147 | https://www.ncbi.nlm.nih.gov/geo/query/acc.cgi?acc=GSE101112 | NCBI Gene Expression Omnibus, GSE101112 |

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
