## [Decision Letter]

**Acceptance summary:**

In this paper the authors characterized two neuroprotective compounds, CMS121 and J147. Although these compounds target different upstream pathways, they are capable of preventing age-associated cognitive decline and metabolic changes in the mouse model. Using transcriptomics and metabolomics, the authors identified that the target pathways converge on the TCA cycle, and acetyl-CoA specifically, demonstrating its causative role in neuroprotection. The study not only enhances the translational value of the two compounds but also provides deeper understanding of processes that are effectively modulated in the course of drug administration. This study is also relevant for the broad audience, because it brings an important step in understanding of age-related pathologies, and provides a new concept to develop prevention strategies and cures.

**Decision letter after peer review:**

Thank you for submitting your article "Elevating acetyl-CoA levels reduces aspects of brain aging" for consideration by *eLife*. Your article has been reviewed by two peer reviewers, and the evaluation has been overseen by a Reviewing Editor and Huda Zoghbi as the Senior Editor. The following individuals involved in review of your submission have agreed to reveal their identity: Lee-way Jin (Reviewer #1); Martin Denzel (Reviewer #2).

The reviewers have discussed the reviews with one another and the Reviewing Editor has drafted this decision to help you prepare a revised submission.

The reviewers find the data valuable and interesting. However, they raised several points that must be addressed in the revised version. As you will see from the reviewers detailed comments appended below, the work requires a number of substantial revisions addressing the concerns. The number of them are to be addressed by additional justifications, explanations or toning down the interpretations. Specifically, the reviewers require to explain and justify an inconsistent experimental design, including the thorough analyses of statistical significance. The reviewers also agree that the manuscript should be experimentally enriched addressing the role of AMPK in mediating these effects. These additional experiments should be performed as thoroughly as possible given the time frame for revisions, i.e. focusing on most important effects and using cultured rat cortical neurons.

Essential revisions:

The data are complex; the readers will benefit from a diagram illustrating key pathways affected by senescence and by CMS121 and J147. Also, the authors state that "the coordinated responses at the gene expression level with aging represent a compensatory mechanism that is linked to dysregulation at the metabolic level". However, the authors only present correlational data and no evidence for this compensatory mechanism.

The description/explanation of the numbers and sexes of mice used is confusing:

1) The authors used only female SAMP8 mice because "women are at a higher risk than men of developing AD". However, this is not exactly a study on AD. On the other hand, when it came to AD mice, only male APPswe/PSEN1dE9 mice were used.

2) "Eleven 9 months old male SAMP8 mice were used as the baseline control group." It is not clear how male "baseline" mice were used (in which experiments) and how their values were compared to female baseline.

3) The Materials and methods indicates 4 mice per treatment group; however, it is indicated in Figure 1, n=5-6/group, and in Figure 2, n=11-18/group. Please clarify. A group number of 4 for behavioral tests is not likely feasible.

4) Subsection “Increases in acetyl-CoA levels are neuroprotective and are associated with inhibition of ACC1 by CMS121 and J147”: there is no Figure 1I, 1J and 1K.

5) Materials and methods: "Because the SAMP8 mice are an inbred strain, the effect of the compounds was only assessed in older SAMP8 mice after the four months of treatment." Please explain.

6) Subsection “CMS121 and J147 enhance acetyl-CoA levels and histone acetylation in neurons”: "acetylation of H3K9 (in APPswe/PSEN1dE9 mice) was increased by the compounds". However, the data and figure legend are confusing. First, comparing Figure 6—figure supplement 1 (D and E) and (F and G) – the Ac-H3K9 level of Tg-AD animals in the latter is much smaller than the former (comparing the green bars), even they appears to be from similar animals. Second, in the legend to Figure 6—figure supplement 1, it is described that tissue in D and E was from 18 months and tissue in F and G was from 23 months. Why was that?

The transcriptome was done in the hippocampus while the metabolome was analysed in the cortex. This and how it affects the analysis of the data need to be explained.

The SAMP8 model needs to be interpreted more carefully regarding normal WT aging. In this context, it would be helpful to compare SAMP8 aging data with published data from WT mice during aging. This would solidify the model. Further, I wonder to what extent the changes in SAMP8 are representative for changes in AD?

The Western blots in Figure 6 show only minor changes. For example, the blot in Figure 6—figure supplement 1 A does not support the change documented in Figure 6C regarding Ac-H3K9. In Figure 6H, pACC1 levels go up with age, while compounds change the signal in the same direction. Thus, these data need to be interpreted more carefully.

The authors have previously identified ATP synthase as a target of J147, with downstream AMPK activation. Is this effect on ATP synthase responsible for the observed effects on the ETC, ACC1 and acetylation? This would be a plausible explanation but is only mentioned in the Introduction. It is therefore essential to experimentally test the role of AMPK in this context. Are the changes caused by the compounds regarding TCA metabolite levels, ACC1 phosphorylation, H3K9 acetylation, and neuroprotection caused by increased AMPK activity?

A color code label is missing for Figure 3 G and H.

The authors state in Figure 3A and B that the majority of the changed genes are the same. It would be great to back this up by showing the overlap. What proportion of these genes are the upregulated fraction?

The text does not properly explain the neurotoxicity assay in Figure 6E.

Generally, the text should follow the order of the figures. In the manuscript, the reader has to jump between sections, making it harder to read. The results in Figures 4, 5, and 6, for example, are presented in a way that is a bit confusing. It would be better to arrange the figures to the order of the text or change the text so that it is not necessary to jump between the figures.

[Editors' note: further revisions were requested prior to acceptance, as described below.]

Thank you for submitting your article "Elevating acetyl-CoA levels reduces aspects of brain aging" for consideration by *eLife*. Your article has been reviewed by two peer reviewers, and the evaluation has been overseen by a Reviewing Editor and Huda Zoghbi as the Senior Editor.

The reviewers have discussed the reviews with one another and the Reviewing Editor has drafted this decision to help you prepare a revised submission.

The reviewers were largely satisfied with the revisions they have requested. However, there is one remaining issue that needs to be addressed experimentally. The authors show that the compounds appear to act through ACC1 inhibition. Inhibition of ACC1 has two consequences: elevated acetyl-CoA and the reduced flux in downstream metabolism. Which of the two effects of ACC1 inhibition is responsible for the observed phenotype should be clarified by experimentally elevating Acetyl-CoA levels by acetate supplementation.

---

## [Author Response]

Essential revisions:The data are complex; the readers will benefit from a diagram illustrating key pathways affected by senescence and by CMS121 and J147.

A diagram summarizing the findings has been added as Figure 7.

Also, the authors state that "the coordinated responses at the gene expression level with aging represent a compensatory mechanism that is linked to dysregulation at the metabolic level". However, the authors only present correlational data and no evidence for this compensatory mechanism.

The sentence has been modified to "We next tested the hypothesis that the coordinated responses at the gene expression level with aging are associated with dysregulation at the metabolic level".

The description/explanation of the numbers and sexes of mice used is confusing:1) The authors used only female SAMP8 mice because "women are at a higher risk than men of developing AD". However, this is not exactly a study on AD. On the other hand, when it came to AD mice, only male APPswe/PSEN1dE9 mice were used.

In order to keep the study affordable and viable, we focused on one gender only. In the past, we have used male mice from the APPswe/PSEN1dE9 strain. Recently, as we started working with the SAMP8 strain, we chose female mice, given that women are at a higher risk than men of developing AD. In order to avoid confusion, we have removed the sentence above from the manuscript.

2) "Eleven 9 months old male SAMP8 mice were used as the baseline control group." It is not clear how male "baseline" mice were used (in which experiments) and how their values were compared to female baseline.

This was a typo and has been corrected to "female SAMP8 mice".

3) The Materials and methods indicates 4 mice per treatment group; however, it is indicated in Figure 1, n=5-6/group, and in Figure 2, n=11-18/group. Please clarify. A group number of 4 for behavioral tests is not likely feasible.

The number of mice in this sentence reflects the mice that died during the course of our study"Six SAMP8 mice fed with control diet, four SAMP8 mice fed with CMS121 diet and four SAMP8 mice fed with J147 diet died throughout the course of this study". The numbers stated in the figure legends are correct.

4) Subsection “Increases in acetyl-CoA levels are neuroprotective and are associated with inhibition of ACC1 by CMS121 and J147”: there is no Figure 1I, 1J and 1K.

This was a typo and has been amended. It should read Figure 6I, 6J and 6K.

5) Materials and methods: "Because the SAMP8 mice are an inbred strain, the effect of the compounds was only assessed in older SAMP8 mice after the four months of treatment." Please explain.

Because we are assessing the effects of our compounds on the aging phenotype, the control group was comprised of 9 months old SAMP8 mice, prior to compound treatment.This sentence has been clarified and now reads "The effect of the compounds was assessed in older SAMP8 mice after the four months of treatment and any age-related changes are defined by comparison to 9 months old SAMP8 mice.".

6) Subsection “CMS121 and J147 enhance acetyl-CoA levels and histone acetylation in neurons”: "acetylation of H3K9 (in APPswe/PSEN1dE9 mice) was increased by the compounds". However, the data and figure legend are confusing. First, comparing Figure 6—figure supplement 1 (D and E) and (F and G) – the Ac-H3K9 level of Tg-AD animals in the latter is much smaller than the former (comparing the green bars), even they appears to be from similar animals. Second, in the legend to Figure 6—figure supplement 1, it is described that tissue in D and E was from 18 months and tissue in F and G was from 23 months. Why was that?

These tissues were generated as part of two separate studies with Tg-AD mice that we had carried out previously. The tissues were already available in our laboratory. Therefore, the differences observed in H3K9 acetylation between mice from the two independent studies might be explained by the age difference. This has been clarified in the Materials and methods and legend of Figure 6—figure supplement 1.

The transcriptome was done in the hippocampus while the metabolome was analysed in the cortex. This and how it affects the analysis of the data need to be explained.

There was not enough tissue to carry out both the transciptomics and metabolomics with the same area of the brain. Based upon data for pathway identification, the global effects of the compounds on these parameters with aging still hold valid despite being from two different parts of the brain, both of which are affected in aging and AD. This is because network analysis using genes and metabolites (GAM) identified acetyl-CoA as a central metabolite regulated by CMS121 and J147 in both cortex and hippocampus, and this finding was further validated in primary neurons, HT22 cells and fibroblasts. In addition, both transcriptomics and metabolomics also identified mitochondrial dysfunction as a feature of aging in SAMP8 mice in both tissues. We have added this information to the Discussion.

The SAMP8 model needs to be interpreted more carefully regarding normal WT aging. In this context, it would be helpful to compare SAMP8 aging data with published data from WT mice during aging. This would solidify the model. Further, I wonder to what extent the changes in SAMP8 are representative for changes in AD?

There are a large number of publications detailing how the phenotype developed by SAMP8 mice with aging is directly relevant to brain aging, and why they should be considered as a model of dementia and sporadic AD (Butterfield et al., 2005; Pallas et al., 2008; Takeda, 2009; Morley et al., 2012; Akiguchi et al., 2017). Some of the pathological traits developed by the SAMP8 mice with aging that are also found in AD include: progressive decline in brain function with early deterioration in learning and memory; increased oxidative stress; inflammation; vascular impairment; gliosis; Aβ accumulation and tau hyperphosphorylation (Takeda, 2009; Morley et al., 2012; Cheng et al., 2014). It is very difficult to compare changes at the molecular level between different strains of mice due to the genetic variability inherent to each strain. There is no ideal mouse model for AD at the moment, but the SAMP8 mice appear to be one of the best to study the overlap of aging and AD. This information and respective references were added and discussed more thoroughly in the revised text.

The Western blots in Figure 6 show only minor changes. For example, the blot in S5A does not support the change documented in Figure 6C regarding Ac-H3K9. In Figure 6H, pACC1 levels go up with age, while compounds change the signal in the same direction. Thus, these data need to be interpreted more carefully.

The data presented in Figure 6C is a result of unbiased band quantification. It should be noted that values of H3K9 acetylation were normalized to total H3 protein, which may be visually misleading. We agree with the reviewers that the data should be interpreted carefully and that is what we tried to do. One possible explanation for the increases in ACC1 phosphorylation in SAMP8 mice with aging could be that this increase represents a compensatory mechanism. This information has now been discussed in the text.

The authors have previously identified ATP synthase as a target of J147, with downstream AMPK activation. Is this effect on ATP synthase responsible for the observed effects on the ETC, ACC1 and acetylation? This would be a plausible explanation but is only mentioned in the Introduction.

In our previous study (Goldberg et al., 2017), we showed that decreasing ATP synthase activity activated AMPK and increased ACC1 phosphorylation. Because ATP synthase has only been identified as the target of J147 and not CMS121, it was not the focus of our study. However, we did carry out experiments to address the role of AMPK in the signaling pathway identified, and these new data were added to the revised text, as detailed next.

It is therefore essential to experimentally test the role of AMPK in this context. Are the changes caused by the compounds regarding TCA metabolite levels, ACC1 phosphorylation, H3K9 acetylation, and neuroprotection caused by increased AMPK activity?

We have now tested the role of AMPK in the mode of action of CMS121 and J147. Using fibroblasts (obtained from Ruben Shaw at Salk Institute) that do not express AMPK (knock out) we show that, similar to what we had found with J147 (Goldberg et al., 2017), the absence of AMPK also reduces the protection against oxytosis by CMS121. In addition, we show that the absence of AMPK prevents the phosphorylation (inhibition) of ACC1 as well as the consequent increases in acetyl-CoA levels. These new data are described in the text and appended as Figure 6—figure supplement 2. Altogether, these results support our conclusion that ACC1 may be a novel target for AD therapy.

A color code label is missing for Figure 3 G and H.

The label has been added to the figures.

The authors state in Figure 3A and B that the majority of the changed genes are the same. It would be great to back this up by showing the overlap. What proportion of these genes are the upregulated fraction?

A diagram showing the overlap of genes was added to Figure 3B. In addition, this information was made clearer in the text, "The top KEGG pathway targeted by both aging and CMS121 was oxidative phosphorylation (OP). […] Surprisingly, the expression of all these genes went up between 9 and 13 months in the SAMP8 mice and this was prevented by CMS121 and J147 (Figure 3C).". The figure legend was also altered accordingly.

The text does not properly explain the neurotoxicity assay in Figure 6E.

This was made clearer in the text, "The primary phenotypic screening model used in our laboratory to identify CMS121 and J147 (Chen et al., 2011; Chiruta et al., 2012) (Figure 6E) relies on the lethal induction of mitochondrial ROS in the mouse hippocampal neuronal cell line HT22 (Prior et al., 2014), a programmed cell death pathway called oxytosis with physiological features similar to those implicated in the nerve cell damage seen in aging and AD (Currais and Maher, 2013; Tan, Schubert, and Maher, 2001). Oxytosis can be triggered by glutamate which inhibits cystine uptake via system X_c_^-^, leading to depletion of intracellular glutathione (GSH), production of ROS and cell death (Tan et al., 2001)".

Generally, the text should follow the order of the figures. In the manuscript, the reader has to jump between sections, making it harder to read. The results in Figures 4, 5, and 6, for example, are presented in a way that is a bit confusing. It would be better to arrange the figures to the order of the text or change the text so that it is not necessary to jump between the figures.

The text in the Results section has been reorganized to match the order of the respective figures. Some graphs in Figure 6 have also been rearranged to follow their description in the text.

[Editors' note: further revisions were requested prior to acceptance, as described below.]

The reviewers were largely satisfied with the revisions they have requested. However, there is one remaining issue that needs to be addressed experimentally. The authors show that the compounds appear to act through ACC1 inhibition. Inhibition of ACC1 has two consequences: elevated Ac-CoA and the reduced flux in downstream metabolism. Which of the two effects of ACC1 inhibition is responsible for the observed phenotype should be clarified by experimentally elevating Acetyl-CoA levels by acetate supplementation.

As per your suggestion, we have now experimentally elevated acetyl-CoA levels by supplementing cells with acetate, a cellular precursor of acetyl-CoA (Pietrocola F., et al. Cell Metabolism, 2015). This was protective in our model, as shown in Figure 6R, supporting our overall conclusions. These new data have been added to the manuscript.